# Colorectal Cancer: Current Updates and Future Perspectives

**DOI:** 10.3390/jcm13010040

**Published:** 2023-12-21

**Authors:** Rosa Marcellinaro, Domenico Spoletini, Michele Grieco, Pasquale Avella, Micaela Cappuccio, Raffaele Troiano, Giorgio Lisi, Giovanni M. Garbarino, Massimo Carlini

**Affiliations:** 1Department of General Surgery, S. Eugenio Hospital, 00144 Rome, Italy; domenico.spoletini@gmail.com (D.S.); dr.griecomichele@gmail.com (M.G.); raf-28@hotmail.it (R.T.); giolimas06@yahoo.it (G.L.); maxcarlini@tiscali.it (M.C.); 2Department of Clinical Medicine and Surgery, University of Naples “Federico II”, 80138 Naples, Italy; pasqualeavella97@gmail.com (P.A.); micaelacappuccio24@gmail.com (M.C.); 3Hepatobiliary and Pancreatic Surgery Unit, Pineta Grande Hospital, Castel Volturno, 81030 Caserta, Italy

**Keywords:** colorectal cancer, incidence, mortality, screening, survival

## Abstract

Colorectal cancer is a frequent neoplasm in western countries, mainly due to dietary and behavioral factors. Its incidence is growing in developing countries for the westernization of foods and lifestyles. An increased incidence rate is observed in patients under 45 years of age. In recent years, the mortality for CRC is decreased, but this trend is slowing. The mortality rate is reducing in those countries where prevention and treatments have been implemented. The survival is increased to over 65%. This trend reflects earlier detection of CRC through routine clinical examinations and screening, more accurate staging through advances in imaging, improvements in surgical techniques, and advances in chemotherapy and radiation. The most important predictor of survival is the stage at diagnosis. The screening programs are able to reduce incidence and mortality rates of CRC. The aim of this paper is to provide a comprehensive overview of incidence, mortality, and survival rate for CRC.

## 1. Introduction

Colorectal cancer (CRC) is the third most commonly diagnosed malignant neoplasm and the second cause of death due to cancer worldwide [1,2,3,4]. There are significant variations in CRC incidence and mortality rates among different countries of the world which are based on different factors: gender [5], age [6,7,8], and ethnicity [6]. It imposes a considerable global load in terms of its complications, mortality, side effects of treatment, health care services utilization, and medical costs [5,9]. The higher rates of incidence are registered in most developed countries, but the diffusion of Western behaviors and lifestyle is responsible for the increase of colorectal cancer cases in less developed countries [10]. The different possibilities of access to care still play a fundamental role in survival and mortality for this neoplasm [11].

In recent years, artificial intelligence has also played a key role in early diagnosis and prediction of cancers [12,13,14,15,16,17,18], including CRC [19,20], as described by numerous studies in the literature. In this clinical context, knowledge of epidemiological data is of great importance to analyze the trend of incidence and prevalence of pathology and to develop new predictive oncological techniques.

Nowadays, the most commonly used CRC serum biomarker is carcinoembryonic antigen (CEA) [21,22,23]. CEA is a protein that is produced by the developing fetus and by some types of cancer cells. CEA levels are typically low in healthy people, but they can be elevated in patients with CRC. CEA is not a specific biomarker for CRC, as it can also be elevated in patients with other diseases [24], such as inflammation of the colon. However, it is a useful biomarker for monitoring patients with CRC for recurrence or progression of the disease [25]. Another CRC serum biomarker is carbohydrate antigen 19-9 (CA 19-9) [22,26,27]. CA 19-9 is a protein that is produced by the pancreas and by some types of cancer cells, including cholangiocarcinoma [28,29,30,31]. CA 19-9 levels are typically low in healthy people, but they can be elevated in patients with CRC, especially those with advanced disease [22,32]. CA 19-9 is not a specific biomarker for CRC, as it can also be elevated in patients with other diseases, such as pancreatitis. However, it can be a useful biomarker for monitoring patients with CRC for recurrence or progression of the disease. SEPT9 [33], methylated SEPT9 [33,34], microRNAs, and circulating tumor DNA (ctDNA), [35] and urine-based biomarkers, as volatile organic compounds (VOCs) [36], microRNAs [37], and DNA methylation markers, showed promising results in early-stage CRC detection.

The aim of this paper is to provide a comprehensive overview of incidence, mortality and survival rate for CRC. Furthermore, we analyze the current screening program available and the future perspectives of serological tests to early detect CRC.

## 2. Biological Background

It is currently estimated that 75–80% of CRCs are sporadic cancers due to the successive accumulation of mutations in genes involved in growth, differentiation, and proliferation of epithelial cells [1]. This multistep carcinogenesis mechanism, called *adenoma–carcinoma sequence*, is due to mutations of at least 15 cancer-related genes. Considered from a molecular perspective, CRC is not a unique pathology, but can be classified into different subtypes characterized by specific genetic and morphological alterations. Chromosomal instability (CIN) is the most common genetic alteration that accounts for 84% of all sporadic CRC; it is characterized by many changes in chromosome numbers and their mutations like deletions, gains, translocations, loss of heterozygosity for a specific genomic regions, and other chromosomal rearrangements. These show a frequent alteration of the number of DNA somatic copies, which are a feature of most tumors that originate by the adenoma–carcinoma sequence. Another subtype of CRC (around ~13–16% of sporadic CRC) is hypermutated and shows microsatellite instability (MSI); this alteration is related to defects in DNA mismatch repair (MMR) system, often associated with wild-type TP53 and a near-diploid pattern of chromosomal instability [1,33]. In addition to the CIN and MSI mechanism, in almost 15% of CRC is evident a hypermethylation of CpG islands at gene promoters with consequent epigenetic silencing of the adjacent genes. This modification of the normal DNA methylation pattern is defined as CpG islands hypermethylation phenotype (CIMP) and contributes to the global deregulation of the expression of genes involved in cell differentiation. Moreover, in almost 40% of sporadic CRC are described mutation in KRAS, HRAS, and NRAS which are responsible for the signal transduction from different growth factor receptors; other common alterations involve PI3K (15–25% of cases) and BRAF (5–10%). Inactivating mutations are also very frequent and the most common involve PTEN phosphatase (10% of cases), TGF-β signaling pathway, and p53 (70% of CRC) [6].

Hereditary forms contribute to about 15–30% of all colorectal cancers [38]. In these conditions, important tumor suppressor or DNA repair genes are silenced by mono allelic gene expression in the germline, and a somatic event (second hit) abolishing the functionality of the residual wildtype allele causes the carcinogenesis. Among the hereditary colorectal cancers, the most common forms are hereditary non-polyposis colon cancer (Lynch syndrome, LS) and familial adenomatous polyposis coli (FAP). Both syndromes are autosomal dominant disorders. Lynch syndrome-associated cancers show signs of mismatch repair deficiency and a faster adenoma–carcinoma transition which takes 3–5 years compared to about 20 years of sporadic CRC. LS also confer an increased risk for extra-colonic cancers such as those of the gynecological, gastrointestinal, hepatobiliary, urological, and nervous system [1,5]. FAP accounts for 1% of all CRCs and its distribution is the same for men and women; it is characterized by hundreds to thousands of adenoma developing in the colon and the rectum due to the germline mutation in adenomatous polyposis coli (APC) gene and shows the classic adenoma–carcinoma sequence. Polyps generally rise in the early teens and lifetime risk of CRC reaches up to 100% if prophylactic colectomy is not performed. Moreover, there is a small potential risk for extra-colonic cancers including that of the duodenum, thyroid, hepatoblastoma, osteomas, stomach, pancreas, and desmoid tumors [1,5].

## 3. Incidence

CRC is the third most commonly diagnosed malignancy after breast and lung cancer with more than 1.9 million of new cases; 72% of these develop in the colon and only 28% originate in the rectum. It accounts for about 10% of all cancer incidence worldwide [39,40]. It is estimated that the incidence of CRC can increase by 60% in 2030 [41], and the patients affected by this neoplasm reach 3.2 million by 2040 [42]. Based on the sex, it is the third most common cancer in men after lung and prostate cancers, and the second most frequent in females after lung cancer [38]. In general, the CRC is more frequent in males than in females. Furthermore, CRC develops in different sites, depending on the sex of the patient: females have cancer in the right colon while males have it in the left colon. Additionally, males have a greater tendency to develop metastatic cancer of the colon while females are more likely to develop metastatic rectal cancer as they age [43].

The global incidence of CRC is not uniformly distributed among regions of the world but varies substantially up to 8- and 6-fold for colon and rectal cancers, respectively [39].

The highest CRC rate is documented in Asia, where are documented 52.3% of all global CCR in 2020; China alone accounts for 28.8% of CRC cases worldwide [43]. Considering rectal cancer, the East Asian regions have the highest incidence rate, particularly Korean males and Macedonian females ranked first [39]. In Europe the incidence rate is 26.9% of all global cases of CCR in 2020 with the highest age-standardized incidence rate (ASIR) reported for Hungary (45.3 per 100,000). The ASIRs in most European countries exceeded 40 per 100,000—higher than the world average rate [44]. Norway ranks first for CRC in females while Hungary ranks first for CRC cases reported in males [45]. In Italy, CCR accounts for 12.7% of all cancers with 48.576 new cases diagnosed in 2020 (25,588 males and 22,988 females). The Italian incidence rate is decreasing in all regions, for both males and females [38]. The incidence of CCR in the United States accounts for 25.6 per 100,000 persons. Colon and rectal cancers incidences are low in Africa and Southern Asia [39,45].

Colorectal cancer can be considered an index of socio-economic development, and its incidence rates tend to rise uniformly with increasing human development index (HDI). The HDI is a statistical index composed of three variables: life expectancy, education (mean years of schooling completed and expected years of schooling upon entering the education system), and per capita income indicators. A country scores a higher level of HDI when its population lives long, is highly educated, and has a high per capita income. Countries undergoing economic growth and westernization (medium HDI nations, such as Brazil, Russia, China, Latin America, the Philippines, and the Baltics) are experiencing increasing incidence of CRC. This trend reflects changes in lifestyle factors and diet: the economic development is responsible of increased consumption of red meat, fat, sugar, animal-source foods, and energy-dense food, which is associated with reduced physical activity and rising of being overweight and obese [46,47,48,49,50]. Most high-HDI nations (such as Canada, the UK, Denmark, and Singapore) have seen an increase in incidence but lowering in mortality, probably due to improved therapies. Highest HDI nations such as the US, Iceland, Japan, and France have witnessed a reduction in both mortality and incidence due to improvement in prevention and treatment [39].

However, the decline in CRC incidence slowed from 3–4% annually during the 2000s to 1% annually during 2011–2019, due partly to an increase in patients younger than 55 years of 1–2% annually since the mid-1990s [38]. The early incidence of CCR in the United States has increased approximately to 45% in adult ages 20–49 years, from 8.6 per 100,000 in 1992 to 13.1 per 100,000 in 2016 [51]. A similar trend is evident among the populations of New Zealand, Australia, Canada, and Northern-Central Europe [43], but not in Italy [52]. Incidence in individuals younger than 50 years increased by about 2% per year for rectal cancer vs. 0.5% per year for tumors in the proximal colon. Early onset patients are also more often diagnosed with advanced disease (27% vs. 21% of older patients) [53].

Several studies reported racial disparity in CRC incidence [39,44,46,51,52]. Siegel et al. [53] have demonstrated a different incidence of mortality depending on racial group in the US during the period 2012–2016; the African Americans showed the highest incidence rate (45.7 per 100,000 persons); it was 38.6 per 100,000 people in non-Hispanic Caucasians and 34.1 per 100,000 persons in Hispanics, the lowest incidence rate was registered in Asian Americans/Pacific Islanders (30.0 per 100,000 people). Several studies focused on the racial difference in genetic susceptibly to CRC found no racial disparity, but the difference in incidence among the ethnic groups seems to be linked to inequality in health care access and exposure to risk factors.

## 4. Mortality

CRC is the second cause of death due to cancer with 935,173 deaths estimated worldwide in 2020 [38], which accounted for about 9% of all cancer-related mortality [5]. The mortality rate for CRC seems to be decreasing in recent years, but this trend has slowed from about 4% annually during the early 2000s to about 2% from 2012 through 2020. Although mortality is decreasing in the majority of developed countries, the number of deaths is estimated to increase by 60.0% for colon cancer and 71.5% for rectal cancer until 2035 [54]. Nevertheless, it is important to underline the increase in surgical interventions in frailer elderly affected by concomitant chronic diseases [55,56].

A disparity in CRC mortality rate and trend for gender has been noticed worldwide: the overall mortality rate is 43% higher in men than in women. A further difference in CRC mortality rate is related to age. The mortality rate all over the world is higher in patients of 65 years and older [57]. However, it is evident an increase in mortality rate among younger compared to older population [53]. According to the authors, this trend could be explained by two different factors: on the one hand, there could be an earlier exposure to the known risk factors of the new generations compared to the previous ones, while on the other hand, it is known that neoplasms appearing at a younger age often show greater biological aggressiveness [6]. According to the data, in the US in 2020, 68% of CRC mortality was registered in patients ≥ 65 years old, 25% in the group 50–64 years, and 7% in patients < 49 years old [5]. In Europe, during the period 1990 and 2016, the CRC mortality rate increased by 1.1% in patients of age group 30–39, while in patients of 40–49 years old, the mortality rate diminished by 2.4% between 1990 and 2009, but raised by 1.1% between 2009 and 2016 [5].

The differences in terms of mortality rate are also related to the racial group. In the US, the African American population showed the highest mortality rate (19.0 per 100,000 persons) while the lowest rate was registered in Asian Americans/Pacific Islanders (9.5 per 100,000 persons). The mortality rate in non-Hispanic whites was 13.8 per 100,000 persons and 11.1 per 100,000 persons in Hispanics [53].

CRC mortality rates change globally, with an attenuated pattern compared to that of the incidence. About 60% of all deaths occur in countries with high or very high HDI [39]. The age-standardized rate of CRC mortality per 100,000 people was 27.1 in very high HDI countries compared to 2.75 in low HDI countries, with a direct proportionality between the CRC mortality index and HDI [5]. Decreasing trends were observed in central European countries (Austria, the Czech Republic, and Germany) and in the United States and Canada [5]. Colorectal cancer-related deaths are increasing in countries with low–medium HDI like countries of Eastern Europe, Asia, and South America. The lowest mortality rates were registered in Ecuador whilst the steepest fall in mortality was in Denmark [53]. In Italy, CCR ranks second in terms of mortality after lung cancer; if only rectal cancer is considered, it ranks ninth in terms of mortality while colon cancer alone ranks eleventh. In 2020, there were an estimated 21,789 deaths with a huge prevalence in males. The Southern regions show a higher mortality rate than Northern regions [38].

## 5. Survival

The five-year relative survival rate for CRC increased by 15%, from 50% in the mid-1970s to 65% during 2012–2018. This trend reflects earlier detection of CRC through routine clinical examinations and screening, more accurate staging through advances in imaging, improvements in surgical techniques, and advances in chemotherapy and radiation [5,58,59,60,61,62]. Stage at diagnosis is the most important predictor of survival, with five-year relative survival ranging from 91% for initial disease to 14% for metastatic one. Approximately 10% of survivors live with metastases, 44% of whom were initially diagnosed with early-stage disease. The largest survival incomes are for metastatic rectal cancer, with 30% of patients diagnosed during 2016–2018 surviving three years compared with 25% only a decade earlier [63].

Liver represents the leading metastase site [3,64,65,66,67,68,69,70,71,72]. Up to 25% of patients simultaneously experienced primary tumour and colorectal liver metastases (CRLM) diagnosis [73], while 20% will progress to stage IV. Although the availability of chemotherapy regimen progress in many primary tumours [15], surgical resection is considered the gold standard treatment for CRLM, with a five-year survival rate from 30% to 60% [9,40,74,75,76,77,78,79,80,81]. However, about 80% of patients are affected by unresectable CRLM, due to bilobar multiple liver metastases and/or extrahepatic disease. Nowadays, systemic chemotherapy regimens [64,76,79,82,83,84,85,86,87,88,89] are proposed to convert patients with initially unresectable CRLM to obtaining and improving long-term surgical and oncological outcomes [23,82,90,91,92,93]. Nevertheless, some patients will progress during neoadjuvant chemotherapy with a debated role of liver resection in this subgroup.

Lung metastases occur in 5–15% of CRC patients; if not treated, metastatic CRC carries a very poor prognosis with a five-year survival rate of less than 5% [63]. During the last two decades, new chemotherapeutics agents have been developed, such as irinotecan, oxaliplatin, and monoclonal antibodies against EGFR and VEGF. These allow prolonged progression-free survival and overall survival in metastatic colorectal cancer [21,23].

Stage at diagnosis reflects the different socio-economic status of racial groups. Black patients are most likely to be diagnosed with metastatic disease than the white ones (25% vs. 21%), likely caused by unequal access to care. However, the five-year relative survival rate for localized disease is quite similar (89–91%) among racial and ethnic groups [63].

Men have a slightly lower five-year relative survival rate (64%) compared with females (65%) despite a more favorable tumor site. In particular, 35% of males vs. 44% of females develop tumors in the proximal colon, which have a higher risk of death compared with left-sided cancers independent of histological and molecular characteristics. However, the largest sex difference in five-year survival is also for left-sided tumors at 66% in men vs. 68% in women for distal colon cancer and 67% vs. 70%, respectively, for rectal cancer [63]. Survival of patients with left-sided tumors was overall higher than that of patients with proximal colon cancer. This observation may be explained by a more favourable stage of cancers located in the distal than in the proximal colon, as well by distinct molecular features between subsites [94].

Survival rates for patients with screen-detected cancer were higher than those found for patients with non-screen-detected cancer within each disease stage. This evidence is probably due to higher adherence to therapy and more healthy behaviour of patients undergoing screening tests compared to non-screen-detected cancers, which contributes to the observed disparities in survival, particularly for patients with stage III and IV cancers.

The CRC survival rate varies among geographic regions. CONCORD-3 study [95] shows the survival for CRC in 71 countries. Five-year survival for colon cancer was higher than 70% in Israel, Jordan, Korea, and Australia. Survival was in the range of 50–69% in 26 countries: Mauritius; Costa Rica and Puerto Rico; Canada and the US; Japan, Singapore, and Taiwan; in 17 European countries (Denmark, Finland, Iceland, Ireland, Norway, Sweden, and the UK; Italy, Portugal, Slovenia, and Spain; Austria, Belgium, France, Germany, the Netherlands, and Switzerland); and in New Zealand. As for colon, five-year net survival for rectal cancer varied widely. Survival was higher than 70% in Jordan (73%), Korea (71%), and Australia (71%). Survival was in the range 60–69% in 24 countries: in Canada and the US; in 4 Asian countries, in 17 European countries: (Denmark, Finland, Iceland, Ireland, and Norway; Sweden and the UK; Italy, Portugal, Slovenia, and Spain; Austria; Belgium; France, Germany, the Netherlands, and Switzerland); and in New Zealand [95].

In Italy, the five-year survival for CRC is 62%, and the ten-year survival rate is 58% both in men and women. The southern regions have survival approximately 5–8% lower than in the Centre–North regions [52].

## 6. Screening and Prevention

Several screening modalities for CRC are currently available: a yearly or two-year fecal occult blood tests (FOBT) or fecal immunochemical test (FIT), sigmoidoscopy every 5 years, or colonoscopy every 10 years [96].

The CRC screening program is recommended for people aged between 50 and 75 years. However, in recent years, the American Cancer Society and the United States Preventive Service Task Force (USPSTF) have lowered the recommended start age for screening in average-risk populations to 45 years old due to the early onset of CRC [5]. Among the European countries The Netherlands reported the highest participation rate of 71.3% to the screening programs, twelve countries reported participation rates of over 50%, while the participation rates in Poland (16.7%) and Belgium (4.5%) were less than 20%. In general, the participation to the screening programs is higher in Northern Europe than in South and Central Europe. In Italy, the participation rate amounts to 45.7%, with higher adherence in the northern regions than in the southern ones.

Over the last decades, screening programs have deeply influenced the incidence and mortality rates of CRC. In the US, a CRC incidence has been reduced by 20% for annual FOBT screening and by 18% for biennial regimen [44]. In Japan, a 60% risk reduction in CRC incidence was achieved among subjects who underwent FOBT screening in comparison to the non-screened population [44]. In the UK, a randomized trial reported that flexible sigmoidoscopy screening resulted in a 26% reduction of CRC incidence [44]. With the implementation of a screening program in Italy, a reduction of cumulative incidence by 10% among the people aged between 50–69 years of age has been registered [44].

In US, FOBT caused reduction in mortality by 32% for the yearly screening and an 18% reduction thanks to the two-yearly screening. FOBT screening caused CRC mortality reduction by 15% in the UK, by 18% in Denmark, and by 16% in France and Sweden [5], by 30% in Japan, and by 31.7% in China [5]. In Italy, the effects of the screening program are similar to other European countries with a reduction of 13% in CRC mortality.

Screening colonoscopy, in addition to diagnosing early lesions or polyps, allows their excision. Complete polypectomy is essential to reduce the risk of early recurrence and the development of interval cancers, defined as the occurrence of CRC following a colonoscopy prior to the next scheduled surveillance procedure. The European society of gastrointestinal endoscopy (ESGE) clinical guideline strongly recommends that all polyps must be resected, except for diminutive rectal and recto-sigmoid polyps which can be predicted with high confidence to be hyperplastic [97,98]. Moreover, adenoma removal significantly reduced the risk of death from colorectal cancer, as compared with that in the general population. Zauber et al. recorded a 53% reduction in mortality from colorectal cancer in patients undergoing polypectomy compared to a control group corresponding [99].

Magnification endoscopy and chromoendoscopy represent useful techniques that enhance lesion demarcation improving adenoma resections. It is linked with a higher sensitivity rate when compared to classical endoscopy [98]. Moreover, the number of artificial intelligence (AI) systems developed or in development for gastrointestinal endoscopy has grown exponentially in recent years. Computer-aided detection (CADe) systems recognise characteristic features in order to discern the presence of a polyp within a still image or a video. Computer-aided diagnosis (CADx) systems are able to distinguish between polyp types and degrees of dysplasia, from benign hyperplastic polyps to advanced cancers, providing a real-time diagnosis to the proceduralist. AI’s ability to assist in precise polyp characterisation can help mitigate the risk of misdiagnoses, reducing unnecessary treatments, and enhancing patient care quality [100].

Despite surgical and endoscopic innovations, many procedures have been proposed but not included in routine clinical examinations: virtual colonoscopy, serum proteomics, and molecular blood tests represent promising tools for the early detection of colorectal lesions. For example, CIMP is already evident in early polyps and in the US is available as an assay able to detect the presence of methylated CpG upon PCR amplification of promoter regions of specific genes starting from DNA extracted from stool or plasma [33,97]. Although these tests are still being evaluated, the sensitivity is 83% and the specificity is 82%; also, this sensitivity is the same for stage I to III of CRC [97]. Further studies are needed to evaluate the sensitivity and specificity of these techniques, but it is reasonable to hope that these tests can overcome current invasive screening methods.

Several risk factors have been identified in the last decades as possible tumorigenesis causes: tobacco use, physical inactivity, obesity, and alcohol.

Even though clinical trials with dietary interventions (e.g., increases in fibre, fruits, and vegetables, and reductions in fat and alcohol) have shown little effect, several observational studies support a role of dietary modifications. Many drugs are being investigated for chemoprevention of this cancer [97,98,101,102]. Several drugs (e.g., aspirin and nonsteroidal anti-inflammatory drugs) are responsible for a significant risk reduction for colorectal cancer or adenomas, but the role of chemoprevention needs to be further defined.

Surgical prevention is established for FAP and ulcerative colitis and the surgical procedure recommended as a gold standard therapy for these patients is restorative proctocolectomy with ileoanal J-pouch [102,103,104,105]. For HNPCC, the role of prophylactic surgery is less well defined, but some authors suggest prophylactic colectomy [102,103]. Because prophylactic surgery is mostly on young, apparently healthy people, morbidity and mortality from surgery has to be kept to a minimum.

## 7. Conclusions

CRC is one of the cancers whose incidence and mortality are modifiable by following healthy lifestyles. However, the burden of CRC is expected to increase due to the aging of the population and to the westernization of less developed countries. Screening programs are able to reduce incidence and mortality rates of CRC. More studies are required to explain the reasons for the increasing burden of CRC in young adults. More efforts are needed to implement screening programs and to control risk factors of CRC to reduce its burden.

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
