# Peer review of "Colorectal Cancer: Current Updates and Future Perspectives"

_jcm, 2023, doi:10.3390/jcm13010040_

Round 1
Reviewer 1 Report
Comments and Suggestions for Authors
1. Differences in colorectal cancer incidence according to each country are mainly discussed; however, there seems to be too much information. It would be better to summarize the regional differences more concisely.
2. Although regional differences in colorectal cancer incidence are mentioned, the discussion about its biological background was not enough.
3. Colorectal cancer risk reduction with polypectomy was not mentioned. Please consider discussing this matter using pivotal studies (Zauber, et al. N Eng J Med. 2012; Sano, et al. Clin Gastroenterol Hepatol. 2023).
4. There are several hereditary tumors associated with colorectal cancer development, and they are known to be strong risk factors. However, they were not mentioned in this manuscript.
Comments on the Quality of English Language
1. It seems that the paragraphs were too separated.
2. In the Methods section, it should be clearly stated what HDI is.
Author Response
I'm very honorated for your comments. My initial idea was to produce an epidemiology work updated with the most recent data, howevwer I tried to follow your advice: a paragraph of biological background and hereditary tumors have been added, the importance of polypectomy is also inserted.
Thank you for giving my article your attention
Reviewer 2 Report
Comments and Suggestions for Authors
R. Marcellinaro and collaborators conducted a comprehensive review of the epidemiology of colorectal cancer (CRC). Their primary objective was to delve into the incidence, mortality, and survival rates associated with CRC. Additionally, they assessed the existing screening programs and looked ahead to the potential of serological tests for early CRC detection. The entire document is written in clear English. It would be beneficial if the authors highlight this review's unique aspects and novelty compared to previous works. Moreover, a more complete discussion on the future of serological testing for early CRC detection would be appreciated. Are there upcoming serological tests that could replace FOBT or FIT?
Author Response
Thank you for giving my article your attention, I'm happy that you appreciated my work
I tried to follow your advice.
Reviewer 3 Report
Comments and Suggestions for Authors
I was very happy to go through your study as this is dealing with a specific topic of my field of expertise and experience. I understand that your primary aim was to contribute to the research by a concise assessment, prioritising the given data in order to make an easy to follow article. Said that I noted a few points to clarify:
wouldn’t be necessary a distinct reference to the rectal cancer? Notably mixing the data for both colon and rectal cancers provides equivocal results
comes following the previous question the sentence stating that the typical metastases are the liver mets. The rectal cancer metastasises to the lungs.
based on the reference n 38 you reported a high incidence rate for the “Macedonian females”. Even if this is a very famous and well known article I wasn’t able to spot your statement
Off these queries I believe it’s a noteworthy paper
Author Response
I'm very honorated to receive your advices.
It's difficult to separate the epidemiological data of colon and rectum because many work consider these neoplasm together.
The metastases to the lung are added.
the citation n38 is an article of global cancer statistic that report the epidemiological data of each cancer. This data show the highest incidence of CRC among the Korean for the male sex and the Macedonian for the female sex in East Asian regions.
Thank you for giving your attention to my work.